# Statins in Cancer Prevention and Therapy

**DOI:** 10.3390/cancers15153948

**Published:** 2023-08-03

**Authors:** Natalia Ricco, Stephen J. Kron

**Affiliations:** 1Basic Sciences Department, Faculty of Medicine and Health Sciences, Universitat Internacional de Catalunya, 08195 Barcelona, Spain; anricco@uic.es; 2Department of Molecular Genetics and Cell Biology and Ludwig Center for Metastasis Research, The University of Chicago, Chicago, IL 60637, USA

**Keywords:** head and neck cancer, statin, cholesterol, prenylation, cancer therapy, prevention

## Abstract

**Simple Summary:**

Retrospective studies of common malignancies such as head and neck cancer often report lower incidence and/or better outcomes for patients incidentally treated with statins, the HMG-CoA reductase inhibitors commonly prescribed to reduce blood cholesterol and related cardiovascular risks. Lipophilic statins have been proposed to both sensitize to therapy and spare normal tissue, suggesting particular benefits in head and neck cancer, where treatment often incurs major toxicities. While roles for statins in prevention remain controversial, rigorous laboratory studies have confirmed the direct effects of statins on cells and tumors that enhance response to chemotherapy, radiation, targeted agents and immunotherapy. This review surveys the literature on mechanisms of action and features of tumors that may mediate the benefits of statins during and following treatment for head and neck cancer. Statins may have their greatest impact on radiotherapy, suggesting prospective studies of prolonged treatment in selected patients toward the long-term goal of treatment de-intensification.

**Abstract:**

Statins, a class of HMG-CoA reductase inhibitors best known for their cholesterol-reducing and cardiovascular protective activity, have also demonstrated promise in cancer prevention and treatment. This review focuses on their potential applications in head and neck cancer (HNC), a common malignancy for which established treatment often fails despite incurring debilitating adverse effects. Preclinical and clinical studies have suggested that statins may enhance HNC sensitivity to radiation and other conventional therapies while protecting normal tissue, but the underlying mechanisms remain poorly defined, likely involving both cholesterol-dependent and -independent effects on diverse cancer-related pathways. This review brings together recent discoveries concerning the anticancer activity of statins relevant to HNC, highlighting their anti-inflammatory activity and impacts on DNA-damage response. We also explore molecular targets and mechanisms and discuss the potential to integrate statins into conventional HNC treatment regimens to improve patient outcomes.

## 1. Introduction

Cancer remains a global health challenge, accounting for some 10 million deaths per year worldwide. Despite substantial advancements in prevention and treatment, cancer remains one of the most common causes of mortality, particularly in older adults [1]. Head and neck cancer (HNC) is a common malignancy that typically presents as a squamous cell carcinoma (HNSCC) originating in the upper aerodigestive tract [2], with over a half million new diagnoses each year worldwide [1]. HNC risk factors vary by site and may encompass tobacco and alcohol use, human papillomavirus (HPV) infection and occupational and environmental hazards [3]. Treatment options include surgery, radiation therapy, chemotherapy, immunotherapy and targeted therapies. Typically, surgery and/or radiation are combined with systemic therapies, depending on the stage and characteristics of the cancer [4]. As with many cancers, HNC diagnosis at an early stage and appropriate treatment selection are associated with complete response and high overall survival, though significant acute and long-term toxicities affect quality of life [5]. Nonetheless, many patients are diagnosed at an advanced stage and/or recur after their initial therapy, the majority of whom will eventually succumb to their disease. There is a critical, unmet need for more effective primary therapies that can be integrated into current treatment regimens without increasing adverse effects. Competitive inhibitors of the mevalonate pathway enzyme 3-hydroxy-3-methylglutaryl-CoA (HMG-CoA) reductase, commonly known as statins, may represent an opportunity to both enhance treatment efficacy and lower toxicity [6,7,8].

By inhibiting HMG-CoA reductase, statins can directly lower cholesterol levels as a result of decreased flux in the mevalonate pathway and reduced formation of precursors for cholesterol biosynthesis (Figure 1). However, the mevalonate pathway is also limiting for production of isoprenoids, which serve key roles in cellular metabolism and signaling such as isopentenyl diphosphate (IPP), farnesyl diphosphate (FPP) and geranylgeranyl diphosphate (GGPP). Statins’ effects on serum cholesterol are both direct and indirect. Decreased intracellular cholesterol levels can induce upregulation of the sterol regulatory element-binding protein (SREBP). This transcription factor governs various genes involved in lipid metabolism, leading to an increased presence of low-density lipoprotein (LDL) receptors on the cell surface. This upregulation enhances the absorption of cholesterol-rich LDL particles from the bloodstream, subsequently lowering plasma LDL-cholesterol levels. Some statins also elevate high-density lipoprotein (HDL), or “good cholesterol”, potentially decreasing the risk of cardiovascular disease [9].

Cholesterol plays vital roles in biological processes including cell membrane maintenance, steroid hormone synthesis, vitamin D and bile acid production and the formation of lipid rafts and caveolae that facilitate transport, signal transduction and cell polarization. In addition, statins’ impact on isoprenoid biosynthesis can produce diverse pleiotropic effects. Isoprenoids serve as lipid anchors for intracellular signaling proteins such as Ras, Rac and Rho, essential for cell growth, survival and differentiation. Modulating isoprenylation can influence cellular signaling pathways, potentially contributing to statins’ therapeutic benefits [7,8].

The cellular impacts of statins are influenced by their chemical structures. Compared to hydrophilic statins such as pravastatin and rosuvastatin that primarily suppress cholesterol biosynthesis by the liver, the lipophilic statins, such as simvastatin, lovastatin and atorvastatin, are more bioavailable in the periphery, providing the potential to impact multiple targets of mevalonate metabolism in cancer cells [10,11]. Simvastatin, a semisynthetic derivative of lovastatin, is the most commonly used statin among the greater than one quarter of U.S. adults over 40 years of age prescribed statins to reduce serum cholesterol levels and/or prevent cardiovascular disease [12]. As such, incidental use of lipophilic statins is a common finding in any large study of cancer patients.

Population-based studies have yielded provocative findings that patients on statins may display lower cancer rates and benefit from more favorable outcomes for cancer treatment, implicating lower serum cholesterol or other effects of these agents in limiting carcinogenesis and resistance [13,14,15,16]. Although specific mechanisms are still debated, there is compelling evidence that statins can sensitize HNC to radiotherapy, chemotherapy and immunotherapy while reducing adverse effects on normal tissue [6,7,8]. Rather than limit the benefits to patients prescribed statins for other reasons, these and other results support studies to evaluate treating HNC patients with statins alongside their cancer treatment toward improving therapeutic outcomes.

This review provides a synopsis of current research on the use of statins in cancer therapy, focusing on head and neck cancer. We conclude that while statins appear very promising as new agents for the treatment of HNC, there remains an urgent need to establish the value of these drugs by pursuing clinical studies beyond impacts of incidental use.

## 2. Cholesterol-Dependent Effects of Statins

Most discussion of cholesterol focuses on the deleterious effects of excess biosynthesis and/or dietary uptake such as atherosclerosis [17]. Nonetheless, cholesterol is a critical component of cell membranes, representing about half of the lipid in plasma membranes, where its content is closely regulated to maintain membrane fluidity and other essential properties in both normal and malignant cells [18]. Further, in cholesterol-rich membranes, cholesterol may partition into rafts and form complexes with specific proteins [19]. As such, decreased cholesterol levels upon inhibition of HMG-CoA reductase by statins would be expected to compromise basic cellular functions. However, where cholesterol starvation differentially impacts cancer cells, this may provide a route to increasing the therapeutic index of cancer therapy.

### 2.1. Membrane Rafts and Signal Transduction

Given the essential roles for cholesterol in maintaining tumor cell membrane functionality, strategies aimed at inhibiting cholesterol biosynthesis could restrict tumor growth and metastasis [20]. As an example, the caveolae protein caveolin-1 (CAV1) has been found to modulate the metastatic and invasive capabilities of oral squamous cell carcinoma (OSCC) cells [21]. Elevated CAV1 expression in metastatic lymph nodes correlates with poor OSCC prognosis. The link to mevalonate pathway metabolism and statins may come via a critical role for cholesterol in lipid raft-dependent functions in cancer cells and tumors [22]. Lipid rafts function as hubs for signaling proteins, selectively and dynamically regulating their recruitment or exclusion in response to intracellular and extracellular stimuli [23]. Via concentrating raft-associated proteins and maintaining stable complexes, lipid rafts facilitate signal transduction, a function mediated in part by CAV1. By disrupting rafts and caveolae, statins can have indirect effects on CAV1 and other proteins, leading to CAV1 degradation and altered cell signaling. Of particular significance, lipid raft integrity modulates survival and cell death pathways [24], suggesting a relationship between lipid rafts and therapy resistance [25]. Indeed, multiple cancer cell survival and proliferation-related signaling pathways have been linked with lipid rafts [26]. Providing a potential mechanism for the beneficial effects of statins, disrupting lipid rafts results in the inhibition of the PIK3/Akt signaling pathway, leading to radiosensitization in HNSCC [27]. A complementary effect may be to limit Akt-induced PD-L1 expression, allowing statins to potentiate anti-tumor immune response [28].

### 2.2. Cholesterol’s Influence on Cancer Cell Proliferation, Survival and Therapy Resistance

Cholesterol’s role in lipid rafts and caveolae in the plasma membrane may impact cancer cell proliferation and survival via effects on signaling by receptors such as HER2 [29], EGFR [30] and CXCR4 [31], as well as transducers and effectors such as PI3K [32], SRC family kinases [25] and NOX [33], along with other regulators [34]. However, maintaining cholesterol at normal levels in other cellular membranes and subdomains impacts a wide range of pathways important to cancer cell growth, proliferation and resistance. Studies of cholesterol depletion by statins or cyclodextrins suggest that cholesterol helps maintain secretory pathway function, regulates autophagy and supports mitochondrial oxidative phosphorylation [17,35,36]. Other connections appear more indirect. ATAD3A, a protein associated with a wide range of physiological and pathological responses, plays a role in cholesterol metabolism [37]. Elevated ATAD3A expression has been observed in various cancers, including HNSCC [38,39,40]. In HNSCC, ATAD3A operates as a mitochondrial oncoprotein that stimulates disease progression via the activation of mitochondrial ERK1/2 [41,42,43]. Notably, the ATAD3A-ERK1/2 signaling pathway links to voltage-dependent anion channel 1 (VDAC1) [41]. VDAC1 promotes the transport of ERK1/2 to the mitochondria, vital for the formation of the ATAD3A-ERK1/2 protein complex in HNSCC cells. Thus, multiple mechanisms may link cholesterol levels to EGFR/PI3K/Akt/mTOR signaling in HNSCC and other cancers [44,45,46].

The calcium-activated chloride channel TMEM16A, previously ANO1, is upregulated in diverse cancers [47] and commonly overexpressed in HNSCC, which is associated with poor outcomes [48,49], establishing it as a therapeutic target. Along with binding to EGFR that may impact HNC proliferation, survival and expression of PD-L1 and thus immune evasion [47,50], TMEM16A has been implicated in resistance to conventional therapy and EGFR-targeted agents. TMEM16A upregulation can also suppress apoptosis and promote cisplatin resistance [51]. Simvastatin impairs TMEM16A channel function—potentially due to cholesterol depletion, though mevalonate-independent mechanisms may be involved—and reduces OSCC cell proliferation in a TMEM16A-dependent manner [49], suggesting statins as an alternative to TMEM16A inhibitors.

### 2.3. Inflammation and Immune Response Modulation

The FDA has approved the anti-PD-1 immune checkpoint blockade (ICB) antibodies nivolumab and pembrolizumab for cisplatin-resistant, relapsed or metastatic HNSCC patients. Clinical trials and research studies have confirmed pembrolizumab’s efficacy and safety, both as a standalone treatment and combined with chemotherapy for recurrent or metastatic HNSCC [52,53]. For patients with PD-L1-positive, relapsed, or metastatic HNSCC, pembrolizumab monotherapy is recommended as a first-line treatment [54]. Despite its potential benefits, ICB therapy faces multiple barriers including intrinsic and acquired resistance and high rates of immune-related adverse events (irAEs). There is considerable interest in combination therapies.

Statins have long been appreciated for their ability to reduce inflammation [55], and part of this effect may be linked to reducing cellular cholesterol levels. Much like its effects on cancer cells, cholesterol depletion may disrupt rafts and caveolae in immune cells, dispersing receptors and transducers that mediate inflammatory signaling in innate and adaptive immune cells [56,57]. As a consequence, a concern would be that lowering cholesterol with statins might reinforce immunosuppression in the tumor microenvironment. Nonetheless, an emerging theme from recent preclinical and patient studies in multiple cancers is that statins potentiate anti-tumor immune responses and/or immunotherapy (e.g., [58,59,60,61]). Multiple cholesterol-dependent mechanisms may be involved.

Nucleic acid detection in the cytoplasm, which activates the innate immune system, occurs through pattern recognition receptors (PRRs). One such PRR is the cyclic GMP-AMP synthase (cGAS)/stimulator of interferon genes (STING) pathway [62]. Activation of cGAS by cytosolic DNA induces STING to phosphorylate and activate TBK1 and drive Type I interferon (IFN) pathway activation, which has the potential to induce an effective anti-tumor immune response [63]. As such, STING agonists are currently being evaluated in multiple contexts as cancer therapeutics [64]. Along with other effects, Type I IFN signaling may limit cholesterol synthesis, which may then further activate STING via depletion of the ER membrane cholesterol pool [65]. Simply disturbing cholesterol metabolism as with statins might be sufficient to induce this positive feedback loop.

Other benefits of statins may be to reduce the suppressive influence of excess cholesterol on immune function. Class II major histocompatibility complex (MHC II) molecules are raft-associated proteins expressed on antigen presenting cells such as dendritic cells (DCs) and serve a key role in presenting processed tumor antigen peptides to effector cells, thereby eliciting anti-tumor responses [66]. Tumor cells can downregulate DC functionality by raising cholesterol levels [67]. Oxysterol secretion by tumor cells impairs DC migration to lymph nodes and reduces T cell priming [68]. 

Cholesterol may also contribute to T cell dysfunction directly via immune checkpoint activation and CD8^+^ T cell exhaustion. Membrane cholesterol serves a direct role in stabilizing PD-L1 through its interaction with cholesterol-binding CRAC motifs [69]. This suggests that reducing cholesterol might be sufficient to interrupt immune checkpoint signaling and/or potentiate immune checkpoint blockade immunotherapy. High cholesterol exposure in the tumor microenvironment is also associated with elevated PD-1 expression by infiltrating CD8^+^ T cells [70] and CD8^+^ T cell exhaustion [71]. 

Multiple indirect effects of statins can be ascribed to the decreased cholesterol biosynthesis. Intracellular cholesterol depletion may promote cleavage and nuclear localization of sterol regulatory element (SRE)-binding proteins (SREBPs [72]) that bind SREs, leading to compensatory expression of sterol pathway and coregulated genes. Along with inducing expression of HMGCR and other mevalonate pathway enzymes, SREBPs increase the expression of enzymes for cholesterol biosynthesis (via SREBP-2) and fatty acid and triglyceride biosynthesis (via SREBP-1) as well as the LDL receptor and diverse other proteins linked to lipid metabolism and transport. Insofar as SREBPs are cancer targets [73,74], this effect of statins may well be counterproductive beyond simply restoring mevalonate pathway activity. Along these lines, one of the SREBP-dependent proteins induced by statins is the LDL receptor negative regulatory factor PCSK9 [75,76]. PCSK9 has emerged as an alternate target for lipid lowering therapy, insofar as inhibiting PCSK9 with antibodies (alirocumab, evolocumab) or siRNA (inclisiran) leads to increased LDLR recycling rather than degradation and greater liver uptake of LDL, lowering circulating cholesterol [77]. Significantly, PCSK9 has a similar effect on MHC I, leading to its lysosomal transport and downregulation [78]. Targeting PCSK9 increased tumor cell MHC I expression, promoted CD8^+^ T cell tumor infiltration and cytotoxicity and potentiated the effects of PD-1/PD-L1 checkpoint blockade. Effects on CD8^+^ T cells may also be direct, as blocking PCSK9 can enhance T cell receptor (TCR) signaling via stabilizing LDLR [79]. While arguing for targeting PCSK9 in cancer immunotherapy [80], these considerations also raise the concern that statins have the potential to interfere with immune checkpoint blockade via activation of SREBP and upregulation of PCSK9.

## 3. Non-Canonical Effects of Statins

Statins appear to exert pleiotropic effects independent of their lipid-lowering properties that may underlie some of their beneficial effects on cardiovascular disease, inflammation and cancer (Figure 2) [81]. Beyond lowering cholesterol, suppressing mevalonate biosynthesis limits formation of isoprenoids and thus reduces protein farnesylation and geranylgeranylation, affecting small GTPases and other modified proteins. Similarly, statins restrict synthesis of ubiquinone (coenzyme Q10) and heme A in cytochrome C oxidase, potentially impairing mitochondrial electron transport chain function. In turn, statins may mediate some of their effects independently of mevalonate pathway inhibition. Physiological consequences of reduced protein prenylation and/or off-target activities constitute non-canonical effects of statins and can have significant impacts on cancer cells.

### 3.1. Anti-Proliferative Effect

Statins have shown growth inhibitory effects on multiple human tumor cell lines including glioma, neuroblastoma, lung and breast cancer cells that appear to be independent of cellular cholesterol but can be partially reversed by FPP and GGPP, indicating the critical role of prenylation [82]. Like Ras and other related small GTPases [83], Rho proteins (Rho, Rac, Cdc42) are prenylated, enabling their membrane association, where they function as conformational switches, transitioning between a GDP-bound inactive state and a GTP-bound active state. Activation of Rho proteins and their effector molecules regulate multiple cancer-relevant pathways including cytoskeletal organization, vesicle trafficking, gene expression, cell signaling, cell cycle, motility and cell survival, supporting tumor initiation, growth, metastasis and therapy resistance [84]. Thereby, the broad reliance on Rho proteins and their requirement for prenylation may serve as an Achilles heel that has the potential to enhance response to therapy or even directly promote cancer cell death such as via activating the intrinsic apoptosis pathway [82]. Indeed, prenylation defects may underlie statins enhancing responses to a wide range of anticancer drugs including EGFR targeted agents [85,86,87]. 

Regarding Rho protein’s roles in proliferation and survival, in the context of HNC, in vitro experiments have demonstrated that atorvastatin inhibits RhoC function, reducing ERK1/2 and STAT3 phosphorylation. This results in reduced cell motility, invasion and colony formation in HNSCC cell lines [88]. In addition, simvastatin has been shown to downregulate integrin beta-1, inhibit stress fiber formation and suppress cell proliferation. Furthermore, simvastatin upregulates cell cycle regulators p21 and p27 [89]. Reduction in tongue squamous cancer cell proliferation and growth were observed upon silencing RhoA, ascribed to altered cyclin D levels and increased p21 and p27 [90]. Statins have been found to stimulate membrane FasL expression and lymphocyte apoptosis via the RhoA/Rho-associated protein kinase (ROCK) pathway in murine melanoma cells [91]. Lovastatin has similarly been observed to inhibit cell proliferation and induce apoptosis in human breast carcinoma cells, potentially through upregulation of p21 and downregulation of cyclin D1 and survivin levels [92]. Treatment with lovastatin or simvastatin in prostate cancer led to RhoA inactivation, inducing cancer cell apoptosis and causing cell cycle arrest in the G1 phase [93]. Considering the pro-proliferative and anti-apoptotic effects of RhoA mediated by activation of ERK1/2, as previously reported [94,95], statin treatment might be expected to reduce ERK1/2 and mTOR phosphorylation levels. This would result in an increase in Bim expression, leading to apoptosis [96]. Supporting this concept, statin-induced apoptosis has been confirmed in both human colon cancer cells and xenografts, as well as in breast cancer [97,98]. Another effector in HNSCC may be the p21-activated kinase PAK2, which is an effector of Rho GTPases involved in chromatin remodeling, cell proliferation and apoptosis. PAK2 overexpression has been positively correlated with chemoresistance and linked to adverse clinical outcomes in HNSCC patients. Upregulation of c-Myc expression by PAK2, leading to transcriptional activation and induction of pyruvate kinase M2 (PKM2) expression, has been identified as resulting in the diminished aerobic glycolysis, proliferation and chemotherapeutic resistance of HNSCC cells [99].

Rac1, a multifunctional protein in endothelial cells, plays various roles including cellular differentiation, adhesion, angiogenesis, migration, vascular permeability and redox signaling. Prenylation is essential for Rac1′s proper subcellular localization [100]. The primary causes of mortality in HNSCC are local invasion and distant metastasis of cancer cells. Persistent Rac1 activation has been reported in HNSCC, with the EGFR/Vav2 axis implicated in cell invasion [101]. Recent work suggested a potential collaboration between Rac1 and the CCL2-CCR4 axis in promoting cell migration and cancer metastasis, emphasizing the potential significance of statins in reducing Rac1 prenylation and thereby function toward enhancing HNSCC treatment outcomes [102]. 

HNSCC is characterized by dysregulation of the autophagy–lysosome pathway, specifically the p62/SQSTM1 protein and overexpression of fibronectin 1 (FN1), which has been associated with poorer prognosis and higher tumor pathological grade in HNSCC patients. Lovastatin is suggested to suppress the expression of FN1 and Rho family small GTP-binding proteins, which are known to contribute to tumor aggressiveness through promoting cellular plasticity [103]. This finding is further supported by another study emphasizing the significant role of statins in regulating fibronectin expression [104]. Collectively, these findings suggest a potential therapeutic role of statins in cancer treatment mediated by impaired function of Rho GTPases.

Some connections to Rho protein prenylation may be less direct. A critical role for the mevalonate pathway, mediated by Rho GTPases, has been established for the mechanosensitive transcription coactivators YAP and TAZ [105], whose activity is mediated by TEAD transcription factors. Surveys of cancer-associated mutations have highlighted YAP and TAZ as oncogenic drivers in a range of malignancies, including HNSCC [106]. Multiple pathways impact YAP/TAZ signaling and thereby limit expression of its targets which include cell proliferation, growth, stress response, immune evasion and survival genes that if deregulated, may help drive cancer initiation, growth, dissemination and therapy resistance [107,108,109,110]. Of particular relevance to ICB immunotherapy, YAP/TAZ regulates expression of PD-L1 in a statin-dependent manner [111,112]. As such, YAP/TAZ and TEAD have emerged as important cancer targets [110]. Blocking Rho prenylation may have multiple effects leading to reduced YAP/TAZ nuclear localization [105] and thus impaired transcription regulation. Statins have also been found to modulate YAP via a mechanism depending on the long noncoding RNA SNGH29 [111].

Given the high incidence of mutations deregulating YAP/TAZ in HNC [113], an attractive model is that beneficial effects of statins in HNC may be mediated by reduced YAP/TAZ activity. Indeed, a genome-wide CRISPR-Cas9-based inactivation screen in OSCC cells led to identifying YAP or TAZ dependency in the majority of HNSCC cell lines [114], validating targeting this pathway. Suggesting an opportunity to combine statins with other agents, YAP1 collaborates with the targetable epigenetic regulator BRD4 [115] to regulate the chromatin accessibility of many genes, which influences their expression and contributes to the malignant properties of HNSCC [116]. 

### 3.2. Chemotherapy Sensitization

While HNSCC is increasingly likely to be treated with ICB immunotherapy, most patients are initially treated with multimodality therapy including surgery, radiation and chemotherapy. Chemoradiotherapy may combine multiple genotoxic agents, such as cisplatin or carboplatin and 5-fluorouracil, along with a targeted agent such as the anti-EGFR antibody cetuximab [117]. Added to radiation, systemic therapy improves local control and overall survival; however, many patients experience severe adverse events during treatment and debilitating long-term toxicities, raising interest in de-escalation, particularly for HPV^+^ disease [118]. Statins may offer the opportunity to reduce the intensity of treatment and/or eliminate the need for radiation or chemotherapy, given their potential to improve the benefits and reduce toxicity [119]. Indeed, several retrospective studies have reported that incidental statin use is associated with improved outcomes in HNSCC [120,121,122]. 

A caveat is that any apparent beneficial effects of combining statins and chemotherapy agents may depend on secondary effects of prolonged statin therapy and/or specific features of each tumor insofar as randomized controlled trials examining adding statins to chemotherapy in several cancers have failed to reveal improved outcomes [6]. In vitro, combining statins with chemotherapy agents has yielded divergent results from sensitization to protection. In a study using breast cancer cells, the combination of simvastatin and doxorubicin increased the amount of cancer cell death when compared to treatment with doxorubicin alone [123]. On the other hand, a different result may arise from the protective effect on healthy cells and tissues, which can mitigate the toxic side effects of chemotherapy. Recently, it has been reported that atorvastatin can reduce intestinal epithelial damage caused by 5-fluorouracil, leading to increased therapeutic index in a mouse model [124].

Unfortunately, mechanistic insights into how statins might impact DNA repair and thereby sensitize HNSCC to chemotherapy are generally lacking beyond the expectation that benefits may be linked to mitochondrial dysfunction and altered tolerance for oxidative stress or due to reduced protein prenylation and function [8]. RhoB is a likely target as it is DNA damage-inducible and enhances DNA double strand break (DSB) repair via activating PP2A in cells treated with camptothecin [125,126]. While knockdown or deletion of RhoB slows DNA repair and induces genomic instability [127], increased expression and accumulation of unprenylated RhoB upon statin treatment may or may not have a similar impact. The synergy of fluvastatin with temozolomide in glioblastoma is ascribed to reduced prenylation limiting Ras activation [128,129]. Regarding small cell lung cancer (SCLC), several human chemoresistant xenograft models exposed to long-term intermittent chemotherapy displayed improved outcomes upon statin treatment. The study revealed that statins can induce oxidative stress and apoptosis via the GGPP synthase-RAB7A-autophagy axis. Furthermore, a negative correlation was observed between GGPS1 expression and patient survival. Notably, combined statin and chemotherapy treatment has resulted in prolonged responses in relapsed SCLC patients who had previously undergone chemotherapy [130]. 

### 3.3. Radiation Sensitization

Like chemotherapy, radiotherapy produces chromosomal DNA damage as part of its therapeutic effects, but its localized delivery is advantageous in potentially concentrating damage in tumors while sparing normal tissue. Unfortunately, the anatomy of the upper aerodigestive tract and the surrounding sensitive tissues of the head and neck confound efforts to limit exposure of normal tissues to toxic radiation doses. Intrinsic and acquired radiotherapy resistance also presents a significant challenge [131]. Co-treatment with chemotherapy to sensitize HNC to radiotherapy exacerbates adverse effects as genotoxic agents can magnify radiation effects in both malignant and normal tissue [132]. As an alternative, we have suggested the mevalonate pathway as a potential target for augmenting radiation sensitivity in HNC without incurring increased normal tissue toxicity [133].

Radiosensitive and radioresistant cell line analysis revealed that the dysregulated mevalonate pathway activity was linked not only to radioresistance but also to increased statin sensitivity. This correlation was substantiated by a decrease in proliferation and viability, along with increased senescence, which aided in sensitizing these cells to ionizing radiation [133]. Examining a series of HNSCC patients undergoing radiotherapy as primary treatment, incidental statin use was found to be significantly associated with improved local tumor control but not on distant sites. While this finding might suggest statins can induce direct sensitization of tumor cells to radiation, there is no consistent pattern in the preclinical literature regarding enhanced effects of ionizing radiation following HmG-CoA reductase inhibition. Our work [134] showed that statin treatment, resulting in loss of protein prenylation, delays repair of chromosomal DSBs following radiation exposure. A repair defect might be indirectly linked to coenzyme Q10 and heme A depletion and mitochondrial dysfunction. However, multiple prenylated proteins may be critical for DSB repair, with Rho GTPases as prominent candidates based on repair defects observed in RhoA and RhoB deficient cells [135]. In pancreatic cancer, activation of a RhoA/ROCK2–YAP/TAZ signaling pathway may mediate radiation resistance [136], potentially inducing expression of DNA damage repair factors. In glioma cells, RhoA-dependent responses to irradiation such as activating DNA damage signaling, formation of ionizing radiation induced foci and DSB repair all depended on wildtype p53 [137]. A potential mechanism may be that RhoA activation increases actin dynamics, releasing G actin that binds p53 and promotes nuclear localization to accelerate DSB repair. 

A consideration here is that while p53 is often wildtype in HPV^+^ HNSCC, mutant p53 is typical for tobacco and alcohol abuse-related cancer and is also linked to radiation resistance [138,139]. As such, p53-independent mechanisms are likely to determine much of the radiosensitization by statins that has been observed. One such mechanism might be via a pathway whereby lovastatin directly binds to and activates the tyrosine phosphatase SHP2 [140,141]. Oncogenic functions of SHP2 have long driven development of targeted inhibitors [142], but among the consequences of SHP2 activation is the dephosphorylation of poly-ADP ribose polymerase 1 (PARP1) [141], thereby reversing the c-Met-mediated activating phosphorylation at Tyr907 [143]. By inhibiting PARP1, statins might slow the repair of multiple forms of radiation-induced DNA damage [144], with the potential to preferentially impact tumor tissue [145].

Among factors limiting HNSCC radiosensitivity, recent evidence points to expression of immune checkpoint proteins, limited antigen presentation and T cell dysfunction [146], suggesting concomitant treatment with ICB as a strategy to improve HNSCC radiation response. Nonetheless, multiple clinical trials have failed to offer strong evidence for synergy [147,148,149]. Besides its direct cytotoxic effects, radiation can modulate inflammation in the tumor microenvironment associated with immunogenic cell death, activated lymphocytic infiltrates and increased antigen presentation. One potential mediator of these effects is increased cytosolic DNA. This leads to cGAS-STING pathway activation and type I interferon (IFNα/β) production, which may potentiate CD8^+^ T cell-mediated tumor destruction [150]. Whether combined with genotoxic therapy or provided on their own, STING agonists have the potential to enhance antitumor responses [151] and have demonstrated encouraging activity in HNSCC [152,153,154,155,156,157], including in HPV^+^ tumors where STING may be targeted by the E7 viral protein [158]. A potential role for statins may be via delaying DSB repair, leading to increased cytosolic DNA that can further stimulate cGAS and STING [140,141]. A caveat is that STING-induced IFNα/β can also be sensed by the tumor cells themselves to drive expression of PD-L1 and promote immune evasion [159]. The additional statin effect of limiting YAP/TAZ activity and thereby reducing PD-L1 expression might help offset this autocrine pathway.

### 3.4. Effects of Mevalonate Pathway Inhibition on Cellular Plasticity and Tumor Microenvironment

Cellular plasticity, such as via the epithelial-to-mesenchymal transition (EMT) [160], contributes to the malignancy of cancer cells by facilitating adaptation to stress, local invasion and metastatic spread. Metastatic potential of pancreatic cancer cells is suppressed by fluvastatin in a dose-dependent manner, associated with significant changes in cell morphology [161]. Similar patterns have been observed in prostate cancer cells treated with rosuvastatin [162]. A possible mechanism involves the inhibition of Akt, which plays multiple roles in regulating cytoskeletal remodeling, cell adhesion and EMT [163] and, as noted above, can be targeted by statins via impacts on lipid rafts or other mechanisms [164]. The Akt pathway is commonly activated in HNSCC, making it an attractive target [165]. Lovastatin, in particular, appears to be a promising candidate to be combined with other therapies to achieve tumor sensitization [166]. Complementary effects may be mediated via inhibiting prenylation of the Rho GTPases, which play critical roles in regulating EMT, migration and invasion along with other cancer hallmarks [84,167]. 

The tumor microenvironment, including metabolism and inflammatory signaling, are important determinants of cell plasticity and EMT in cancer cells. As such, multiple effects of statins on the TME may impact tumor resistance, recurrence and metastasis. The combination of celecoxib and simvastatin has been shown to significantly reduce HNSCC proliferation [168]. These effects extend to potentiating anti-tumor immune response. Statins influence TME metabolism in lung cancer and thereby enhance immune checkpoint blockade [169]. Similarly, combining statins with cisplatin creates a TME favorable for immunotherapy in HNSCC [170].

## 4. Protective Effects of Statins

Regarding prevention, while multiple studies have suggested reduced risk of cancer, including HNC with long-term statin use, this is not a strong effect, and a recent case-control study found that prior exposure to statins in HNC patients is not associated with lower cancer risk [171]. Meta-analyses have revealed statin use may contribute to lowering the incidence of specific cancers such as hepatocellular carcinoma (HCC) [172], though umbrella reviews surveying multiple cancer types identified overall weak evidence for benefits on incidence [173] or survival [174], pointing to considerable variability among cancer sites. As such, use of lipophilic statins as cancer-preventative agents may only be relevant to specific high-risk populations. However, there appears to be a stronger argument that continued use of statins during and then after treatment may be beneficial in protecting patients with HNSCC—especially those with HPV^+^ disease—from adverse outcomes of therapy [119,120,121,122].

Chemoradiation therapy is particularly associated with significant acute and late toxicities, which may increase non-cancer-related mortality. Statins may protect against anthracycline cardiotoxicity, but this has yet to be confirmed by RCT [175], and these agents are not typically used in HNC. Statins appear to have a protective effect that can limit inflammation and normal tissue damage after radiation in preclinical models [176], with effects on tissues with direct relevance to HNC including those of the lung [177] and salivary glands [178]. These effects may be mediated by both cholesterol and prenylation-dependent mechanisms.

Statins have shown the ability to alleviate some treatment-related toxicities that can have a detrimental impact on the quality of life among individuals who have survived cancer. Ototoxicity caused by cisplatin appears to be lower with incidental statin use [179]. Pravastatin has been found to reduce the thickness and severity of radiation-induced fibrosis in patients with HNSCC [145]. 

Successfully treated HNC patients remain at high risk of both secondary cancers and non-cancer causes of death [180]. Cardiovascular disease and stroke appear to be particularly increased in treated HNC patients [181]. Although the advanced age and history of smoking typical of HNC patients may be most responsible, there is a specific risk of stroke that has been linked to vascular injury during radiotherapy [182]. Given these considerations, it is not surprising that statins may offer significant protection against the increased rates of stroke and other vascular events after irradiation of the head and neck [183,184]. 

## 5. Conclusions

Head and neck cancer remains a considerable challenge, associated with the difficult anatomy and intrinsic resistance of the tumors to therapy. Statins have long been proposed as safe and effective agents with the potential to improve treatment outcomes for these patients. In particular, blocking mevalonate biosynthesis and thereby limiting protein prenylation appears to have a significant impact on the response to radiation by slowing DNA repair. A recent advance has been to implicate statins in potentiating anti-tumor immune responses. Among the contributing mechanisms, lowering cholesterol levels in the tumor microenvironment may promote antigen presentation and T cell effector function, while reduced protein prenylation may impact DNA repair, and the resulting accumulation of cytosolic DNA may drive cGAS/STING signaling. These and other effects may contribute to an overall benefit from statin treatment.

Given the evidence from retrospective studies and the strong rationale provided by laboratory studies in cells and tumor models, there is a compelling argument for randomized trials of concomitant treatment with lipophilic statins and radiotherapy in head and neck cancer, selecting patients with no history of treatment with statins for cardiovascular disease. To obtain the full benefits of statins, initiating treatment upon cancer diagnosis and then continuing for a prolonged course may be necessary. This may have the dual advantage of increasing the efficacy of radiotherapy but also reducing late side effects such as increased stroke risk. Looking forward, if statins are indeed able to increase the therapeutic ratio of head and neck radiotherapy, they might find their greatest value in reducing the need for genotoxic chemotherapy as a radiosensitizer, thereby facilitating treatment de-intensification, particularly in HPV^+^ patients.

## Figures and Tables

**Figure 1 cancers-15-03948-f001:**
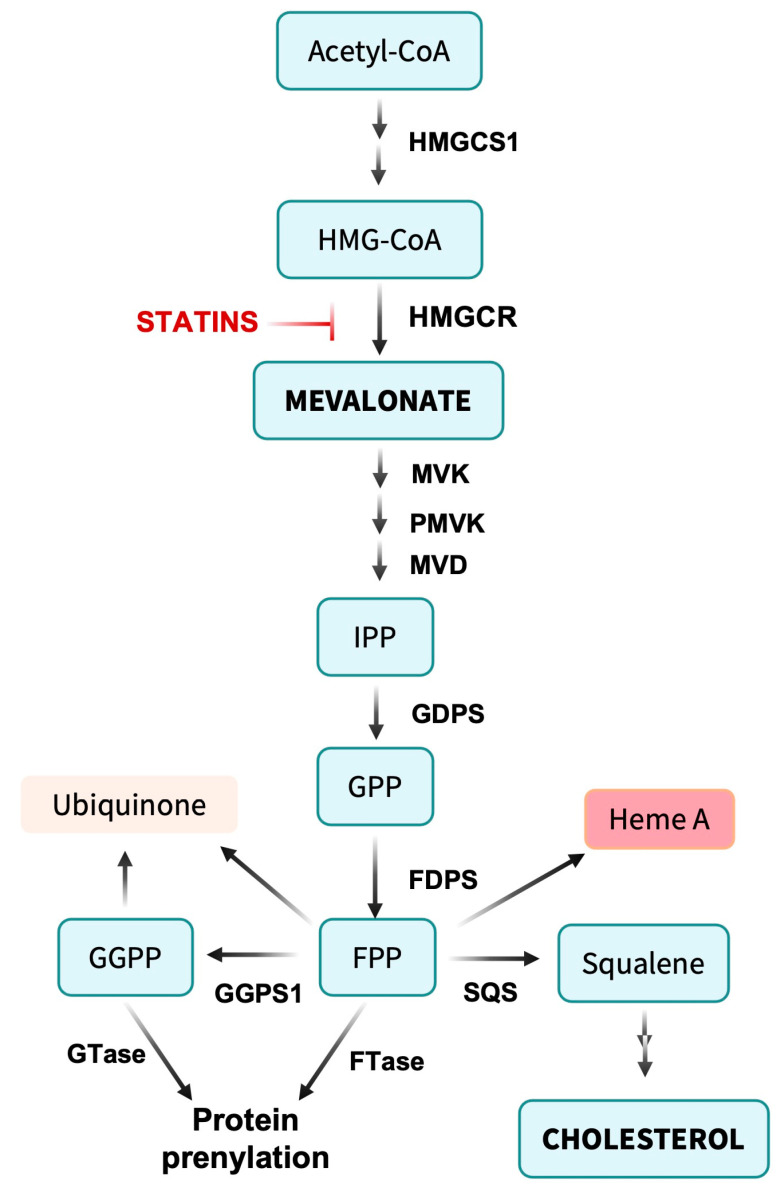
Overview of the mevalonate pathway. Acetyl-CoA is condensed to form 3-hydroxy-3-methylglutaryl-CoA (HMG-CoA). The rate-limiting step in the pathway involves the conversion of HMG-CoA to mevalonate by HMG-CoA reductase (HMGCR). Mevalonate undergoes further metabolism by mevalonate kinase (MVK), phosphomevalonate kinase (PMVK) and mevalonate decarboxylase (MVD) to yield isopentenyl pyrophosphate (IPP). IPP is converted to farnesyl pyrophosphate (FPP) by farnesyl diphosphate synthase (FDPS) and to geranyl geranyl pyrophosphate (GGPP) by geranyl geranyl diphosphate synthase (GGPPS1), providing intermediates for protein prenylation. Alternatively, two FPP molecules are linked by squalene synthase (SQS) to form the C30 isoprenoid squalene, a rate-limiting precursor for cholesterol synthesis. FPP can be directly added to biomolecules during the formation of ubiquinone (coenzyme Q10) and heme A of cytochrome c oxidase, supporting mitochondrial metabolism. Not shown are pathways from IPP to tRNA isopentenylation.

**Figure 2 cancers-15-03948-f002:**
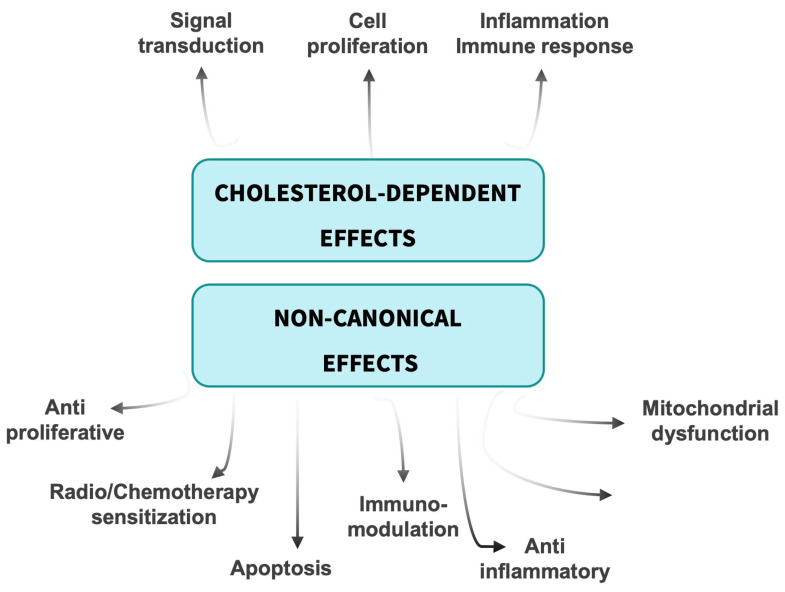
Schematic of cholesterol-dependent and non-canonical effects of statins that may impact response to cancer treatment in head and neck cancer.

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
