# Peer review of "Statins in Cancer Prevention and Therapy"

_cancers, 2023, doi:10.3390/cancers15153948_

Round 1

Reviewer 1 Report

Interesting and well comprehensive review. I have only some minor remarks:

1) the authors described properly the mechanisms of action of statins in this field but the manuscript would benefit from further clinical data. The authors should describe some clinical evidence on this beneficial role (for example cite and comment the role of statins in HCC, PMID: 32260179)

Author Response

We thank the Reviewers for their feedback and the Editor for giving us the opportunity to prepare a revision after the submission. We have revised the manuscript to respond to Reviewer queries and address key statin mechanisms that we had overlooked, adding new topics and additional relevant.

Reviewer 1 comment: Interesting and well comprehensive review.

Response: We thank the Reviewer for their comment.

Point 1: I have only some minor remarks: the authors described properly the mechanisms of action of statins in this field but the manuscript would benefit from further clinical data. The authors should describe some clinical evidence on this beneficial role (for example cite and comment the role of statins in HCC, PMID: 32260179)

Response 1: we concur with the Reviewer and have added a brief discussion of this consideration between L503 and L509.

Reviewer 2 Report

Very interesting, well written review on anticancer activity of statins.

Just a few minor comments

L44 Can you add numbers for HNC and HNSCC

L157 ER2, EGFR, PI3K, NOX, GPCRs, are not in reference 26 please add appropriate references

L347 Needs more discussion and  References

Author Response

We thank the Reviewers for their feedback and the Editor for giving us the opportunity to prepare a revision after the submission. We have revised the manuscript to respond to Reviewer queries and address key statin mechanisms that we had overlooked, adding new topics and additional relevant.

Reviewer 2 comment: Very interesting, well written review on anticancer activity of statins.

Response: we thank the Reviewer for their enthusiasm.

Point 1: L44 Can you add numbers for HNC and HNSCC

Response 1: we concur with the Reviewer and have added the requested information in L44-L45

Point 2: L157 ER2, EGFR, PI3K, NOX, GPCRs, are not in reference 26 please add appropriate references

Response 2: We concur with the Reviewer and have added the references between L161 – L163.

Point 3: L347 Needs more discussion and References

Response 3: we have expanded this section and added citations between L373 and L384.

Reviewer 3 Report

This is a very interesting review describing several Statin-induced mechanisms to improve clinical outcomes in Head and Neck cancer patients under various therapeutic modalities. The review is at many aspects descriptive and could be improved provided the authors would discuss in more details some of these mechanisms. For instance, interference of Statins in signaling pathways (eg inhibition of phosphorylation in the AKT, mTOR, ERK pathway) associated with tumor cell motility inhibition; the impact of concomitant Statin treatment in immunomodulatory pathways (e.g. those associated with tumor-infiltrating lymphocytes and with IL-6 or those linked to mutated PIK3CA and Akt) in patients receiving immunotherapies or targeted therapies. The authors should also discuss Statin-induced tumor microenvironment remodeling via down-regulation of immune suppressor pathways.

none

Author Response

We thank the Reviewers for their feedback and the Editor for giving us the opportunity to prepare a revision after the submission. We have revised the manuscript to respond to Reviewer queries and address key statin mechanisms that we had overlooked, adding new topics and additional relevant references. We hope this revised manuscript is now acceptable for publication.

Comment Reviewer 3: This is a very interesting review describing several Statin-induced mechanisms to improve clinical outcomes in Head and Neck cancer patients under various therapeutic modalities.

Response: We appreciate this comment.

Point 1: The review is at many aspects descriptive and could be improved provided the authors would discuss in more details some of these mechanisms. For instance, interference of Statins in signaling pathways (eg inhibition of phosphorylation in the AKT, mTOR, ERK pathway) associated with tumor cell motility inhibition; the impact of concomitant Statin treatment in immunomodulatory pathways (e.g. those associated with tumor-infiltrating lymphocytes and with IL-6 or those linked to mutated PIK3CA and Akt) in patients receiving immunotherapies or targeted therapies. The authors should also discuss Statin-induced tumor microenvironment remodeling via down-regulation of immune suppressor pathways.

Response 1: we thank the Reviewer for their helpful feedback. We have increased our attention to defining molecular mechanisms throughout. We have added further discussion of how statins may impact signaling by the Akt pathway and other regulators and thereby modulate multiple cancer hallmarks via their effects on cholesterol and/or prenylation. We realized that we had overlooked a few key topics such as mevalonate pathway impacts on mitochondrial electron transport and reactive oxygen and now give them appropriate attention. We also note the potentially deleterious outcomes of activating the SREBP feedback loop (L237-L258). Further, a new section, 2.4 (L471-L494), has been added to the paper highlighting the tumor microenvironment as a statin target to impact cancer cell invasion, motility and sensitivity to immunotherapy. We hope these revisions address the Reviewer's concerns.

Round 2

Reviewer 1 Report

The revised version of the paper is OK. Thank you!